Predominant intragenic methylation is associated with gene expression characteristics in a bivalve mollusc

Gavery Mackenzie R.
Roberts Steven B. sr320@u.washington.edu
School of Aquatic and Fishery Sciences, University of Washington , Seattle, WA , USA
Yang Xiang-Jiao
Electronic publication date: 2013 Nov 21
Publication date: 2013
Volume: 1
Electronic Location ID: e215
Received 2013 Sep 26; Accepted 2013 Nov 1
Copyright: © 2013 Gavery and Roberts
Copyright year: 2013
Copyright holder: Gavery and Roberts
License: This is an open access article distributed under the terms of the Creative Commons Attribution License, which permits unrestricted use, distribution, and reproduction in any medium, provided the original author and source are credited.
License URL: https://creativecommons.org/licenses/by/3.0/

Keywords: DNA methylation, Epigenetics, Bivalve, Mollusc, Gene regulation, Methylome

Funding: Environmental Protection Agency STAR Fellowship Assistance Agreement number FP917331 National Science Foundation Grant Number 1158119 This work was supported in part by the National Science Foundation (NSF) under Grant Number 1158119 awarded to SR Roberts, and the U.S. Environmental Protection Agency (EPA) STAR Fellowship Assistance Agreement no. FP917331 awarded to MR Gavery. Any opinions, findings, and conclusions or recommendations expressed in this material are those of the author(s) and do not necessarily reflect the views of NSF or the EPA. The funders had no role in study design, data collection and analysis, decision to publish, or preparation of the manuscript.

==============================
Characterization of DNA methylation patterns in the Pacific oyster, Crassostrea gigas, indicates that this epigenetic mechanism plays an important functional role in gene regulation and may be involved in the regulation of developmental processes and environmental responses. However, previous studies have been limited to in silico analyses or characterization of DNA methylation at the single gene level. Here, we have employed a genome-wide approach to gain insight into how DNA methylation supports the regulation of the genome in C. gigas. Using a combination of methylation enrichment and high-throughput bisulfite sequencing, we have been able to map methylation at over 2.5 million individual CpG loci. This is the first high-resolution methylome generated for a molluscan species. Results indicate that methylation varies spatially across the genome with a majority of the methylated sites mapping to intra genic regions. The bisulfite sequencing data was combined with RNA-seq data to examine genome-wide relationships between gene body methylation and gene expression, where it was shown that methylated genes are associated with high transcript abundance and low variation in expression between tissue types. The combined data suggest DNA methylation plays a complex role in regulating genome activity in bivalves.

Introduction

Epigenetic marks such as DNA methylation are important for genome regulation (Bell & Felsenfeld, 2000; Li, Bestor & Jaenisch, 1992; Hsieh, 1994). DNA methylation has been well-studied in mammals and plants where it has been shown to play important roles in temporal and spatial regulation of gene expression (Okano et al., 1999; Zhang et al., 2006), reduction of transcriptional noise (Bird, 1995), and genome stabilization (Wolffe & Matzke, 1999). However, the distribution and context of DNA methylation varies greatly among phylogenetic groups indicating that these functional roles may not be conserved (Colot & Rossignol, 1999).

In contrast to the heavily methylated vertebrate genomes, invertebrate genomes exhibit a relatively low level of methylation that is primarily confined to gene bodies (Zemach et al., 2010). It is perhaps because of these differences that DNA methylation has remained largely understudied in invertebrates. Recently, however, there has been a renewed interest in invertebrate DNA methylation patterns as it is now being recognized that invertebrates are exceptional models to study functions and evolutionary roles of gene body methylation. Furthermore, it has been shown that DNA methylation mediates phenotypes in response to environmental cues in the invertebrate Apis mellifera (Kucharski et al., 2008; Lyko et al., 2010), indicating an important role in integrating environmental signals. Thus, understanding both the distribution and function of DNA methylation in diverse invertebrate lineages will contribute significantly to our understanding of the evolution of genome regulation and environmental physiology.

The focus of the work presented here is to explore the role of DNA methylation in bivalve molluscs. The presence of DNA methylation has been confirmed in several bivalve species (Wang et al., 2008; Petrovic et al., 2009; Gavery & Roberts, 2010). A majority of the research on DNA methylation in molluscs has focused on the Pacific oyster (Crassostrea gigas), an economically and ecologically important species. Previous studies in the Pacific oyster identified a relationship between gene function and methylation pattern. Specifically, it was shown that genes with housekeeping functions are more methylated than genes involved in inducible functions (i.e., genes involved in response to environment, embryonic development or tissue-specific functions) (Gavery & Roberts, 2010; Roberts & Gavery, 2012). More recently, Riviere et al. (2013) determined that DNA methylation plays a critical role in development as indicated by differential methylation patterns throughout embryogenesis. This was further supported by their observation that 5-aza-cytidine, a potent demethylating agent, significantly disrupts embryonic development (Riviere et al., 2013).

These recent studies on DNA methylation in oysters provide important foundational information on DNA methylation in bivalves. However, previous studies were not able to provide fine scale resolution of DNA methylation patterns, nor examine the relationship with gene expression at the genome-wide level. Here, we provide the first high resolution methylome of a mollusc and examine this in relationship to gene expression data to get a better understanding of the role of DNA methylation in invertebrates.

Methods

Bisulfite treated DNA (BS-Seq) analysis

The cohort of adult oysters used in this study was from Samish Bay, WA, USA. Briefly, genomic DNA was isolated using DNAzol (Molecular Research Center) from gill tissue of 8 oysters, pooled, and methylation enrichment performed using the MethylMiner Kit (Invitrogen) following the manufacturer’s instructions. Specifically, pooled DNA was sheared by sonication on a Covaris S2 (Covaris) (parameters: 10 cycles at 60 s each, duty cycle of 10%, intensity of 5, 100 cycles/burst). Approximately 13 µg of sheared DNA was used as input DNA and incubated with MBD-Biotin Protein coupled to M-280 Streptavidin Dynabeads following the manufacturer’s instructions (MethylMiner (Invitrogen)). Enriched, methylated DNA was eluted from the bead complex with 1 M NaCl and purified by ethanol precipitation. This enriched fraction represented approximately 15% of the total DNA recovered from the enrichment procedure. The DNA library was prepared using the Illumina Tru-Seq system with methylated TruSeq adapters (mean fragment size of library: 350 bp). Bisulfite treatment was then performed using the EpiTect Bisulfite Kit (Qiagen) following manufacturer instructions. Library preparation and sequencing was performed on the Illumina HiSeq 2000 platform at the University of Washington high throughput sequencing facility (Seattle, WA). High-throughput reads (36 bp single end) were mapped back to the oyster genome (Fang et al., 2012) using BSMAP software version 2.73 (Xi & Li, 2009). Methylation ratios (i.e., number of unconverted cytosines/the number of converted and unconverted cytosines at each locus) were extracted from BSMAP output (SAM) using a Python script (methratio.py) that is distributed with the BSMAP package. Only cytosines in a CpG context with sufficient sequencing depth (defined here as greater than or equal to 5× coverage) were retained for further analysis.

DNA methylation landscape of genomic features

In order to examine relationships between DNA methylation and genomic features, data from BSMAP (i.e., methratio) was converted to genomic feature tracks (i.e., generic feature format [GFF] files). Conversion was done using SQLShare (Howe et al., 2011), with the files and corresponding query language published (Gavery & Roberts, 2013).

The distribution of methylated CpGs with respect to specific genomic features was determined using BEDtools (i.e., intersectBED) (Quinlan & Hall, 2010). For this analysis, a CpG locus was considered methylated if at least half of the reads remained unconverted after bisulfite treatment. Genomic features that were examined include: exons and introns (Fang et al., 2012), putative promoters (defined as 1 kb upstream of open reading frames), and transposable elements. Putative transposable elements were identified using RepeatMasker (Smit, Hubley & Green, 1996–2010), based on protein similarities to the Transposable Element Protein Database. At the time of analysis the database contained 5411 predicted proteins. For comparative purposes, total CpG across the entire C. gigas genome was also examined. Locations of all CpGs were identified using the EMBOSS tool fuzznuc (Rice, Longden & Bleasby, 2000), and the proportion of total CpG in each of the genomic features listed above was determined using intersectBED. A Chi-squared test was performed to determine if the distribution of methylated CpG was different from what would be expected by a random distribution of the total CpG in the genome (p-value < 0.05 was considered significant).

Average methylation ratios were determined for full length genes and also the cumulative exons and cumulative introns comprising a gene. Average methylation was determined by the number of methylated cytosines divided by the total number of CpG per region. The correlation between the methylation status of exons and introns of individual genes was performed using Pearson’s correlation coefficient in SPSS (SPSS Inc.).

The relationship between predicted methylation status, using the CpG observed to expected ratio (CpGo/e), and the average methylation ratio for each gene was examined to assess the effectiveness of the CpGo/e method for predicting methylation in bivalve species. For this analysis, the CpGo/e ratio was calculated for each gene using the method described in Gavery & Roberts (2010). Correlation between CpGo/e and the methylation ratio was performed using Spearman rank correlation in SPSS (SPSS Inc.).

Gene expression analysis

RNA was isolated from gill tissue of the same 8 individuals used for individuals used for bisulfite sequencing using Tri-Reagent (Molecular Research Center). RNA was pooled in equal quantities and enriched for mRNA using Sera-Mag oligo dT beads (Thermo Scientific). First strand synthesis was performed using SuperScript III (Invitrogen) and the second strand of cDNA was synthesized using dUTP instead of dTTP, making the library strand-specific. A shotgun library was constructed from double stranded cDNA for paired end sequencing by end-polishing, A-tailing and ligation of sequencing adaptors. Sequencing was performed on the Illumina HiSeq 2000 platform at the Northwest Genomics Center at the University of Washington (Seattle, WA). High-throughput reads (50 bp paired end) were mapped back to the oyster genome (Fang et al., 2012) using CLC Genomics Workbench version 6.5 (CLC Bio). Initially, sequences were trimmed based on quality scores of 0.05 (Phred; Ewing & Green, 1998; Ewing et al., 1998), and the number of ambiguous nucleotides (>2 on ends). Sequences smaller than 20 bp were also removed. For RNA-Seq analysis, expression values were measured as RPKM (reads per kilobase of exon model per million mapped reads) (Mortazavi et al., 2008) with an unspecific match limit of 10 and maximum number of 2 mismatches.

The RPKM values were used to examine the relationship between gene expression and DNA methylation in gill tissue. All genes containing at least 1 CpG locus (n = 28,105) were grouped into deciles according to transcriptomic representation in gill tissue (RPKM) and the average methylation ratios for each decile were compared. A one-way ANOVA followed by Tukey’s test for multiple comparisons was performed using R (R Development Core Team, 2012) and a significance level of p < 0.05 was accepted.

A principal component analysis (PCA) was used as an exploratory tool to identify relationships between DNA methylation, gene expression profiles and gene attributes such as length. To explore variables related to gene expression, publicly available RNA-seq data from a variety of adult C. gigas tissues were leveraged from Zhang et al. (2012). Specifically, mean transcript abundance and variation in transcript abundance across tissues were calculated using RPKM values for 7 tissues adult tissues (digestive gland, female and male gonad, gill, anterior muscle, hemocytes and labial palps). Mean transcript abundance was calculated using the mean RPKM across all tissues for each gene. Variation in transcript abundance across tissues was calculated as the coefficient of variation (%CV) of the RPKM across all 7 tissues for each gene. Other gene attributes that were examined, as they may associate with DNA methylation, include gene length, number of exons per gene and number of CpG per gene. In summary, the following attributes were included as variables in the PCA performed in R (R Development Core Team, 2012): average methylation ratio of the full length gene (as described above), gene length in base pairs (bp), number of exons, average transcriptomic representation (average RPKM across 7 adult tissues), coefficient of variation (%CV) of transcript abundance (RPKM) among tissues. All variables were log transformed, with the exception of the methylation ratio which was arcsine transformed prior to analysis. The significance of each principal component was calculated using Monte-Carlo randomization tests. Principal components were considered significant at p ≤ 0.05. Correlation loadings of ≥0.6 were considered significant.

Results

DNA methylation mapping

Bisulfite treated DNA sequence reads (139,728,554 total reads; 36 bp) are available in the NCBI Short Read Archive under the accession number SRX32737. A total of 120,734,949 reads (86%) mapped to the C. gigas genome. Fifty-six percent of the 164,873,219 cytosines in the C. gigas genome had at minimum of 1× coverage. Of the 9,978,551 CpG dinucleotides in the genome, 2.6 million (26%) had ≥5× coverage. The distribution of methylation ratios found at CpG dinucleotides ranged between 0.0 and 1.0, but a majority of the loci were either heavily methylated or unmethylated. Specifically, 55% (1,453,752) were methylated (i.e., ≥ 0.50) and another 28% were unmethylated (i.e., = 0.0) (Fig. 1). Genome feature track files (i.e., GFF) representing (1) all CpG dinucleotides and (2) methylated CpG dinucleotides (>50%) for this dataset were developed and are available (Gavery & Roberts, 2013).

Figure 1 Frequency distribution of methylation ratios for CpG dinucleotides in oyster gill tissue.

A total of 2,625,745 CpG dinucleotides with ≥5× coverage are represented.

Methylation landscape of genomic features

Methylated CpG dinucleotides, defined as having a methylation ratio of 0.5 or greater, were located predominantly in intragenic regions (exons and introns), but were also present in putative promoters (defined as 1 kb upstream of TSS), transposable elements and unannotated intergenic regions. The distribution of methylated CpG across various genomic regions is significantly different than what would be expected if the methylation were distributed randomly throughout the CpG dinucleotides in the genome (X2 = 513, 194.1, df = 4, p < 0.0001). Specifically, DNA methylation appears to be overrepresented in intragenic regions (64% of methylated CpG in combined exons and introns) when compared to the proportion of all CpG in the genome (38%) (Fig. 2). When methylation was examined on a per gene basis a strong positive correlation (R2: 0.86) was observed between exonic and intronic methylation within a gene. Additionally, a strong correlation was observed between the gene methylation measured via high-throughput bisulfite sequencing and the predicted methylation ratio based on the CpG observed to expected ratio (CpGo/e) (Spearman rho: −0.616, p-value: <1 × 10−4).

Figure 2 Comparison of the total CpG versus methylated CpG in oyster gill tissue by genomic feature.

Proportion of all CpG (blue) and methylated CpG (red) in gill tissue across genomic features of C. gigas. Percent of CpG dinucleotides in Exons, Introns, Transposable Elements (TE), promoters (Pro) and unannotated intergenic regions (Other) are reported.

Gene expression & DNA methylation

After quality trimming, 45,751,574 RNA-seq reads mapped to the genome. Raw reads are available in the NCBI Short Read Archive under the accession number SRX367081.

The relationship between the proportion of methylation in a gene and overall transcript abundance in gill tissue was characterized (Fig. 3). In general, transcription abundance increases significantly with increasing DNA methylation until the 40th percentile after which it remains relatively stable until the 100th percentile when methylation significantly decreases.

Figure 3 DNA methylation among genes with increasing transcript abundances.

Expressed genes were grouped into deciles by transcription abundance. Genes not expressed in gill (i.e., RPKM = 0) are also shown (leftmost column). Error bars represent 95% confidence intervals.

The first two principal components (PC) of the PCA of gene attributes were significant and explained 76.4% of the variation among the genes. This variation was being driven by multiple factors, including DNA methylation (Fig. 4 and Table 1). The only variable that did not contribute significantly to the first two principal components was mean transcript abundance (correlation loading 0.2). The first PC, which explained 50.2% of the variation was loaded heavily by number of CpG dinucleotides, the length of the mRNA and the number of exons. The second PC, which explained 26.1% of the variation was loaded heavily by the %CV of gene expression among tissues and the methylation ratio. DNA methylation is negatively correlated with transcript variance between tissues (%CV) and relatively uncorrelated with attributes such as gene length.

Figure 4 PCA ordination of oyster genes by gene attributes.

Variables loadings shown by purple arrows. Variables significantly contributing to PC1 and PC2 include: methylation ratio (Methylation), the coefficient of variance of expression between tissues (%CV), the number of exons (Exons), the length of the mRNA in base pairs (mRNA) and the number of CpG dinucleotides in the gene (CG). Variables that did not significantly contributes to PC1 and PC2 include the mean transcript abundance (Expression). Inset depicts ordination of the genes analyzed on PC1 and PC2 (n = 27, 181).

Table 1 Summary of PCA for gene attributes.

Principal
component	%variance
explained	Significance
value	Significant variable loadings	
PC1	50.2	<0.001	Number of CpG	0.9	
Length mRNA	0.9	
Number of exons	0.8	
PC2	26.1	<0.0001	Expression %CV	−0.6	
Methylation ratio	0.6	

Discussion

Here we have used methylation enriched high-throughput bisulfite sequencing in conjunction with genomic feature annotation and transcriptomic data to gain a better understanding of the role of DNA methylation in oysters. This work not only provides new information on DNA methylation in invertebrates but also provides a framework for characterizing DNA methylation in other taxa.

The reduced representation approach was selected to obtain a higher coverage of methylated regions. In addition, since methylation was likely to occur in gene bodies (Zemach et al., 2010), and because transcriptomic data was the primary genomic resource for C. gigas at the time of sequencing (the genome was released soon after), it was expected that methylation enrichment would significantly limit the proportion of unmappable reads. Quantitative methylation data were obtained for both methylated CpG as well as unmethylated CpG that were either interspersed with or flanking these more heavily methylated regions. Therefore, methylation enriched bisulfite-sequencing was effective in generating a comprehensive invertebrate methylome.

One of our primary findings was the overall level of genome methylation in the oyster. Here we found that 15% of CpG dinucleotides (2% of total cytosines) are methylated in gill tissue. This degree of methylation is much lower than the global methylation patterns seen in mammals where 70%–80% of CpGs are methylated (Bird & Taggart, 1980), but still higher than what has been reported in other invertebrates. For instance, only 0.8% of the CpGs are methylated in the brain of A. mellifera (Lyko et al., 2010) and between 0 and 8% of CpGs are methylated in the nematode, Trichinella spiralis, depending on the life stage (Gao et al., 2012). Although methylation in C. gigas is relatively high for an invertebrate, it is not outside the range of what has been reported in other species by liquid chromatography-mass spectrometry analysis. For example, similar to the oyster, 2% of total cytosines are methylated in the mollusc Biomphalaria glabrata (Fneich et al., 2013). It should be noted that methylation in oysters does likely vary in both a temporal and possibly tissue specific manner, as clearly indicated by Riviere et al. (2013) by characterizing differences in total methylation during development. In addition, because the sample represents a pool of multiple individuals, it cannot be determined whether the variation in methylation at a particular locus represents hemimethylation or differential methylation between individuals. In general, the bimodal pattern observed (Fig. 1) indicates that a CpG locus is either heavily methylated or unmethylated, but future work sequencing individual oysters would provide valuable information regarding individual epigenetic variation in oysters.

This work also provided the first direct evidence in oysters that DNA methylation is prominent in gene bodies (see Fig. 2) and these data are well correlated with previous investigations using an in silico approach (i.e., CpGo/e) to predict methylation in C. gigas (Gavery & Roberts, 2010).The predominance of gene body methylation is consistent with what has been described in other invertebrates (e.g., Suzuki et al., 2007; Zemach et al., 2010; Lyko et al., 2010) and there is increasing evidence that gene body methylation is the ancestral pattern (Lechner et al., 2013). The function of gene body methylation remains unclear, but studies indicate possible active roles in preventing spurious transcription (Bird, 1995; Huh et al., 2013) and regulating alternative splicing (Maunakea et al., 2010; Shukla et al., 2011; Foret et al., 2012), as well as a more passive role for methylation as a byproduct of an open chromatin state (Jjingo et al., 2013). Given the nature of the study design, we are not able to directly test the hypothesis that DNA methylation contributes to spurious transcription or the regulation of alternative isoforms in C.gigas. However, genomic feature tracks have been developed and published (Gavery & Roberts, 2013) so that genome wide methylation can be easily visualized with respect to gene expression patterns (exon-specific RPKM).

Exons are the preferential target for gene body methylation for most species (Feng et al., 2010), and methylation is enriched in exons of the oyster. However, there is also a relatively large amount of intronic methylation in oysters when compared to other invertebrate species. For example, DNA methylation occurs almost exclusively in exons in the honey bee A. mellifera (Lyko et al., 2010). Genome-wide methylation studies in other invertebrate species also report very low levels of intronic methylation relative to other genomic regions (e.g., Gao et al., 2012; Bonasio et al., 2012). Similarly, in plants, methylation is preferably targeted to exons; however, it has been reported that in globally methylated mammalian genomes gene body methylation is not biased toward exons (Feng et al., 2010), although exon/intron boundaries can be marked by differences in DNA methylation (Sati et al., 2012). It appears that bivalves may be unique among the invertebrates examined in terms of the degree of methylation in introns. Intronic methylation has been implicated to be involved in gene regulation through the expression of alternative isoforms of genes in other species (e.g., Maunakea et al., 2010; Foret et al., 2012). Variation in methylation patterns between taxa may indicate that additional model invertebrates are needed to study the function of these epigenetic marks.

The distribution of DNA methylation in the C. gigas genome is consistent with the fractionated or ‘mosaic’ pattern of methylation previously described in invertebrates (Tweedie et al., 1999; Simmen et al., 1999). In oysters, as in other invertebrates, the methylated fraction tends to consist of gene bodies, while other genomic regions exhibit less methylation (Fig. 2). Interestingly, transposable elements (TE) show little methylation in oyster gill tissue. This is in contrast to vertebrate genomes where TE are heavily methylated and function to suppress their activity (Yoder, Walsh & Bestor, 1997). While there is no general consensus regarding the extent of TE methylation across invertebrate taxa, the pattern of sparse TE methylation observed in oysters is similar to what has been described in other invertebrate species (Simmen et al., 1999; Feng et al., 2010; Zemach et al., 2010).

Intragenic DNA methylation is positively correlated with gene expression in C. gigas with moderately and highly expressed genes showing the highest degree of methylation (Fig. 3). This relationship is similar to what has been reported for other invertebrate species (Zemach et al., 2010). Interestingly, Riviere et al. (2013) reported a negative relationship between DNA methylation and expression of certain homeobox (hox) genes during embryonic development in C. gigas. The authors hypothesized that the apparent suppression of hox expression by DNA methylation may be due to repression by DNA methylation proximal to the transcription start site in these genes. Although the results reported here and those of Riviere et al. may appear contradictory, it is possible that depending on the context of the methylation (i.e., whether gene body or promoter methylation) it may play either a repressive or expressive role. This is referred to as the DNA methylation paradox (Jones, 1999) and is observed in a wide range of taxa.

We used an ordination approach to explore genomic attributes or groups of attributes that predictably co-occur with methylated genes in the C. gigas genome. Because multiple factors may be linked with methylation (either through causative or correlative associations), this approach allowed us to identify relationships between multiple variables. The most interesting finding from the PCA analysis is that the amount of methylation in a gene is related to the variance in expression between tissues. Genes that show the least variation in expression between tissues have higher DNA methylation levels than those exhibiting a tissue-specific expression profile (i.e., high %CV between tissues). This observation provides corroboration for previous reports based on in silico analyses in oysters showing that housekeeping genes have the highest amount of methylation in C. gigas (Gavery & Roberts, 2010). Housekeeping genes perform functions required by all cell types, therefore it’s expected that their expression patterns would show low variation across tissues. The results of this study are consistent with the expectation that genes with low expression variation across tissues show a high degree of methylation relative to genes with a more tissue-specific expression pattern. Again, this study supports previous findings (Gavery & Roberts, 2010; Roberts & Gavery, 2012) that heavily methylated genes are enriched in housekeeping functions, which are essential for cellular function. One theory is that the lack of methylation in genes with tissue-specific expression can contribute to phenotypic plasticity by allowing more transcriptional opportunities through processes such as allowing access to alternative TSS, facilitating exon skipping or other alternative splicing mechanisms and allowing for increased sequencing variation (Roberts & Gavery, 2012).

Conclusions

Through the current effort, quantitative methylation data were obtained for over 2.5 million CpG dinucleotides throughout the genome of Crassostrea gigas. These data represent the first high resolution methylome in any mollusc and the analytical approaches provide a framework for DNA methylation characterization in other species. In addition, the dataset developed here will be beneficial for phylogenetic analysis of DNA methylation in invertebrates, which will be more robust with the addition of a lophotrochozoan species. The results of this study highlight similarities in epigenetic profiles of invertebrates such as a predominance of gene body methylation and a positive relationship between intragenic methylation and gene expression. In addition, they highlight interesting differences between invertebrate epigenomes including a higher level of intronic methylation in bivalves than what has been reported, for example, in insects. Although the functional role of DNA methylation in bivalves remains elusive, two scenarios could explain our findings. One possibility is DNA methylation in gene bodies is a byproduct of transcription resulting from an open chromatin state, as proposed by Jjingo et al. (2013). Thus the methylation patterns are influenced by transcriptional activity. The second scenario is DNA methylation is involved in regulating gene activity in bivalves. If in fact DNA methylation does influence transcription, the regulatory role is likely very complex. For instance, DNA methylation could have both a have direct regulatory effect on certain genes as proposed by Riviere et al. (2013), as well as facilitating expanded transcriptional opportunities in other cases. Future studies will certainly be challenging given the dynamic nature of DNA methylation, but will hopefully help better delineate if DNA methylation plays a functional role in regulating genome activity in bivalves and what that role might be.

The authors would like to thank Bill Howe and Daniel Halperin for their assistance with SQLShare. They would also like to acknowledge Samuel J. White for comments that improved the manuscript.

Additional Information and Declarations

Competing Interests

Author Contributions

DNA Deposition

Data Deposition

The authors declare no competing interests.

Mackenzie R. Gavery conceived and designed the experiments, performed the experiments, analyzed the data, wrote the paper.

Steven B. Roberts conceived and designed the experiments, analyzed the data, contributed reagents/materials/analysis tools.

The following information was supplied regarding the deposition of DNA sequences:

NCBI Short Read Archive: accession number SRX32737, SRX367081.

The following information was supplied regarding the deposition of related data:

Gavery M, Roberts S. 2013. Crassostrea gigas high-throughput bisulfite sequencing (gill tissue). figshare. [http://dx.doi.org/10.6084/m9.figshare.749728] Retrieved 19:17, Sep 20, 2013 (GMT).

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
