# Peer review of "Predominant intragenic methylation is associated with gene expression characteristics in a bivalve mollusc"

_PeerJ, doi:10.7717/peerj.215_

## Round 0.1 · original submission · Major Revisions

The reviewers' comments are very constructive and I hope that you will be able to revise the manuscript accordingly.

Reviewer 1 ·

Basic reporting

The manuscript entitled “Predominant intragenic methylation is associated with gene expression and tissue specificity in a bivalve mollusc” by Gavery and Roberts describes their analyses of genome-wide relationships between gene body methylation and gene expression. While for the most part I find their analyses to be interesting, it is mostly a descriptive study with no examples of specific genes that provide insight or corroborate their finding.

Experimental design

The authors of this study state that “methylation in oysters does likely vary in both a temporal and possibly tissue specific manner”, while the Zhang’s study observed that “The oyster genome is highly polymorphic”. Given that “The cohort of adult oysters used in this study was from Samish Bay, WA, USA”, while the pacific oyster used in Zhang’s study is an inbred female produced by four generations of brother–sister mating. Question is the compatibility of the methylome data from this study with the RNA-seq data from Zhang’s study. RT-PCR validation of the expression levels of selected example genes and some in-depth discussion would help to address this issue.

Validity of the findings

The author noted that “Specifically, DNA methylation appears to be overrepresented in intragenic regions” and in intron and Intronic methylation has been implicated to be involved in gene regulation through the expression of alternative isoforms of genes. Beside just global analyse of the methylome data, some detailed analyse on example genes would provide some insight.

Additional comments

Missing ")" on line 120.

Reviewer 2 ·

Basic reporting

The authors report a high-resolution methylome analysis in Crassotrea gigas. They used a bisulfite conversion strategy that allows identifying DNA methylation at single nucleotide resolution. They show that methylation is essentially intragenic and associated with gene expression as this was previously demonstrated in other invertebrate species.

Experimental design

They used commercially available kits to enrich their starting biological material in methylated DNA and to do the bisulfite treatment. They used the illumina strategy for the sequencing of their fragmented DNA. Their bioinformatic pipeline is robust and based on software that was previously successfully used.

Validity of the findings

This is the first report of a genome wide methylome analysis in Crassotrea gigas and also in a molluscan species. This work is certainly of great value to help at deciphering methylated region in this organism. Therefore this publication will certainly be a reference for further analysis in oyster and other bivalves. It comes with other papers that were recently published in the field with the aim at describing the structure of methylated DNA in invertebrate species. This is so far a unique reference for the molluscan phyla. I have however some suggestion to improve the quality of the manuscript (please see next section).

Additional comments

Comments on the method section:

About the MethylMiner Kit (Invitrogen) :
- This kit requires DNA to be fragmented before the methylation enrichment. There is no details about the fragmentation of the DNA, starting amount of DNA? Was sonication used, which machine, what condition? what was the size of the fragmented DNA?
- Enriched methylated DNA may be eluted either as a single fraction or doing fractionate steps with increasing NaCl concentration, which condition was used?

About the MethylMiner Kit (Invitrogen) :
- Several procedure exists depending on the starting amount of DNA, salt concentration etc.. Which one was used? What was the starting amount of DNA for this procedure?

In this sense, an accurate summary of each of these two procedures would be appreciated for the readers to be able to repeat the experiment. Please, could you also provide a brief summary of the library construction and sequencing experiment?



Comments on the result section:
The authors published in a previous paper the distribution of predicted methylation status of 12,210 annotated C. gigas transcripts measured computationally by CpGo/e ratio. This distribution is bimodal as observed in other invertebrate organisms. The results presented in the figure 1 of this paper seem to confirm this bimodality. I would find interesting to perform a correlation analysis between in silico predicted methylation status and methylation ratio in genes that encodes these 12,210 transcripts. This kind of correlation has been validated in insect species previously in the work of Sarda et al (2012) and it would be interesting to validate it in mollusk. It would be helpful to predict methylation profil based on in silico prediction in other bivalves.

The principal component analysis is not clear; the authors should revise this section and clearly explain the aim of their analysis and their interpretation.



Comments on the discussion section:
The authors highlight similarities in epigenetic profiles with other invertebrates such as a predominance of gene body methylation and a positive relationship between intragenic methylation and gene expression. Other point could be discussed: transposon methylation and mosaic like feature. Indeed, genomes of invertebrates are characterized by interspaced regions of methylated and unmethylated DNA. Have the authors any evidence for such a mosaic like feature in oyster?


Comments on the title:
I agree with a title mentioning that predominant intragenic methylation is associated with gene expression in a bivalve mollusk, but I don’t see where the authors discussed the association with tissue specificity. The authors should revise their title or go deeper into an analysis that indeed demonstrates a link with tissue specificity.


Minor revision:
L90: Supplementary instead of supplemental
L213: Riviere instead of Riverie
Legend of figure 3, y axis: of is repeated twice

---

## Round 0.2 · accepted · Accept

I agree with the reviewers that this is a nice contribution.

·

Basic reporting

Ok

Experimental design

OK

Validity of the findings

OK

Additional comments

OK

Reviewer 2 ·

Basic reporting

The authors have submitted a revised version of a high-resolution methylome analysis in Crassotrea gigas. the quality of the manuscript has improved, thay have taken into account all my request.

Experimental design

the overall experimental design has not changed.

Validity of the findings

As previously mentionned, This work is certainly of great value to help at deciphering methylated region in this organism. Therefore this publication will certainly be a reference for further analysis in oyster and other bivalves.

Additional comments

The requested work has been performed and has hemped to improve the quality of the manuscript.